# Natural Immunity against HIV-1: Progression of Understanding after Association Studies

**DOI:** 10.3390/v14061243

**Published:** 2022-06-08

**Authors:** Ma Luo

**Affiliations:** 1National Microbiology Laboratory, Public Health Agency of Canada, Winnipeg, MB R3E 3L5, Canada; ma.luo@phac-aspc.gc.ca or ma.luo@umanitoba.ca; 2Department of Medical Microbiology and Infectious Diseases, Rady Faculty of Health Sciences, University of Manitoba, Winnipeg, MB R3E 0J9, Canada

**Keywords:** HLA, the PCS vaccine, FREM1, TILRR, inflammation

## Abstract

Natural immunity against HIV has been observed in many individuals in the world. Among them, a group of female sex workers enrolled in the Pumwani sex worker cohort remained HIV uninfected for more than 30 years despite high-risk sex work. Many studies have been carried out to understand this natural immunity to HIV in the hope to develop effective vaccines and preventions. This review focuses on two such examples. These studies started from identifying immunogenetic or genetic associations with resistance to HIV acquisition, and followed up with an in-depth investigation to understand the biological relevance of the correlations of protection, and to develop and test novel vaccines and preventions.

## 1. Introduction

It is more than 40 years since clinical AIDS was first observed in the United States [1] and nearly 40 years since the virus, human immunodeficiency virus type 1 (HIV), causing AIDS was identified [2,3]. In contrast to the speed and success of the development and evaluation of the vaccines to prevent severe disease from SARS-CoV-2 infection, an effective HIV vaccine to prevent HIV acquisition is still not in sight. The failures of several candidate HIV vaccines that have been in large scale human clinical trials [4,5,6,7,8] disappointed HIV vaccine investigators, founders, and the world. Why have vaccine approaches worked well for other infectious pathogens but failed for HIV? Is it the reason that HIV is too diverse and it mutates rapidly? Is it the reason that HIV integrates into the host genome soon after its infection [9]? Is it the reason that HIV infects CD4 T cells, an important part of the immune system, and activation of them would help the virus to propagate and establish in the host? Studies have shown that all of these characteristics of HIV have made the development of an effective prophylactic vaccine especially challenging; among them, some are more difficult to overcome than others. Instead of comparing data from the previous large scale human vaccine trials to search for answers [9,10,11,12,13], this review will try to find some answers from understanding natural immunity observed in a group of HIV-resistant Kenyan female sex workers.

## 2. Natural Immunity to HIV

Variability in susceptibility to infectious disease has long been observed in human history, either within a population or among different populations [14,15,16]. When exposed to an infectious pathogen, some individuals do not appear to become infected, or respond with mild symptoms and recover very quickly after being infected, while others succumb to the infection. It is recognized that the outcome of infections by infectious pathogens depends on the interactions between the pathogens and host [17,18,19,20]. The diversity of the pathogens and the polymorphism of host genetic factors are all important in determining the result of infection. It is logical to believe that by understanding why some individuals are protected from infection and/or disease, an effective vaccine/prevention or treatment can be developed.

Because HIV infection does not cause AIDS immediately and the progression to AIDS and death without anti-retroviral treatment takes several years, the natural immunity to HIV infection and disease progression could not be readily observed without careful monitoring and a long-term prospective follow-up of populations that are at high risk of HIV infection. The Pumwani sex worker cohort established in Nairobi, Kenya in 1985—the early phase of the HIV pandemic—is one such rare study cohort for HIV. The cohort was initially established as an observational cohort to study the immunobiology and epidemiology of sexually transmitted infections (STI) [21,22,23,24]. It is located in the heart of the Pumwani slum in Nairobi, Kenya, and has continued to recruit participants for 30 years. The cohort enrollees have been followed biannually. In addition to research, it provides services related to STI and HIV prevention and care, including consultation, the provision of free condoms, and treatment of other infections. There was no program for anti-retroviral drug treatment from 1985 to 2003 (PEPFAR was introduced to Kenya in 2003). Through more than 20 years of biannual biological and clinical follow-ups, a subgroup of women who are resistant to HIV infection has been identified [25]. These women remain seronegative and PCR negative for HIV for prolonged periods despite heavy exposure to the virus through active sex work [25,26]. Over the years, many studies have been conducted to understand this natural immunity to HIV in the hope that by understanding this natural immunity, an effective HIV vaccine and/or HIV prevention can be developed [26,27,28,29,30,31,32,33,34,35,36,37,38,39,40,41,42,43,44,45,46,47,48]. This review focuses on the progression of two such studies. These studies started from the identification of genetic or immunogenetic variants enriched among the HIV-resistant female sex workers, and then continued with follow-up studies to understand the biological relevance of the identified associations, and to the development and testing of the novel HIV vaccine candidate and potential therapeutics to prevent HIV infections.

## 3. Human Leukocyte Antigens and the Differential Susceptibility to HIV Infection

Human leukocyte antigen (HLA) class I and class II genes are centrally involved in the host’s adaptive immune responses to infectious pathogens. Together with NK cell receptor genes, HLA class I genes are also involved in the host innate immune response. HLA class I proteins present antigenic peptides of pathogens to CD8+ T cells, which destroy the infected cells together with the pathogens. HLA class II antigens present antigenic epitopes to CD4+ T cells to initiate CD4+ T cell responses. HLA class I and class II genes are very polymorphic, and account for varied host immune responses to many infectious pathogens. The polymorphisms in exons 2 and 3 of class I antigens and exon 2 of class II antigens contribute to their variable binding abilities to different peptides and the spectrum of antigen presentation within populations. By April 2022, 24,308 HLA class I and 9182 class II alleles have been identified in human populations worldwide. The 23,417 alleles of the three classic class I genes, HLA-A (7452), -B (8849) and -C (7393) encode 13,793 variant proteins [49]. The 9073 alleles of nine functional HLA class II genes (DRA, DRB1, DRB3, DRB4, DRB5, DQA1, DQB1, DPA1, and DPB1) encode 5749 variant proteins [49]. The extreme genetic diversity of the HLA genes allows the human immune system to recognize and initiate effective immune responses to a broader range of pathogens at the population level [50,51]. When a human population encounters an infectious pathogen, individuals with HLA class I and II alleles that can recognize critical pathogenic peptides would be able to initiate effective immune responses to eliminate the infectious pathogens and survive. Studies have demonstrated the critical roles of HLA polymorphism in HIV infection and disease progression [52,53,54,55,56,57,58,59,60,61,62,63,64,65].

### 3.1. Identification of HLA Class I Alleles Associated with Resistance/Susceptibility to HIV Infection

HIV-specific CD8+ and CD4+ T cell responses were detected in the HIV-resistant women of the Pumwani cohort [27,28,31,34,36,66]. These HIV-specific T cell responses could play an important role in protecting these women from HIV infection. These HIV-resistant women may have specific HLA class I and/or class II alleles that enable their immune system to recognize a specific part of HIV and make their CD8+ and/or CD4+ T cell responses more effective. The alleles associated with protection would be enriched among the HIV-resistant women, and the alleles associated with increased susceptibility to HIV acquisition would be enriched in HIV-infected women. The Pumwani sex worker cohort was established in 1985, the early phase of the HIV pandemic in Africa, and many HIV-uninfected women seroconverted rapidly after enrolment, while a small but substantial number of them remained uninfected [25,32]. Thus, it is anticipated that by the analysis of HLA class I and class II alleles of these women enrolled in the Pumwani cohort, the alleles associated with resistance or susceptibility to HIV infection could be identified. We comprehensively analyzed HLA class I and class II genes of the cohort enrollees and identified HLA class I and class II alleles associated with differential susceptibility to HIV infection [39,40,41,63,67,68]. Our studies showed that HLA class I alleles A*01, C*06:02, and C*07:01 are significantly enriched in HIV-resistant women, and women with these alleles seroconverted significantly slower than women without these alleles [67], whereas A*23:01, B*07:02, B*42:01, C*02:10, and C*07:02 are associated with increased susceptibility, and women with these alleles were rapidly infected after enrollment [67]. A number of HLA class II alleles associated with differential susceptibility to HIV infection have also been identified [39,40,41]. In this review I focus on the follow-up studies of two HLA class I alleles associated with different outcomes of HIV infection. These two HLA class I alleles are A*01:01 and B*07:02. A*01:01 is enriched in the HIV-resistant women. A*01:01+ woman who enrolled in the cohort HIV negative, seroconverted significantly slower than A*01:01– women [45,67], whereas B*07:02 is enriched in HIV-infected women. B*07:02+ women seroconverted rapidly after cohort enrolment [45,67] (see Figure 1).

### 3.2. Analysis of CD8 T Cell Epitopes of A*01:01 and B*07:02

Because HLA class I alleles associated with different outcomes of HIV infection are most likely due to the differences in their abilities to present HIV epitopes to the CD8+ T cells and the induced immune responses following antigen recognition, analysis of their HIV epitopes might find vital clues for developing an effective HIV vaccine. We used a combination of two approaches to study and compare the HIV CD8+ T cell epitopes of A*01:01 and B*07:02. First, we systemically screened overlapping peptides of HIV Gag proteins in vitro using iTopia Epitope Discovery System (Beckman Coulter) [45]. The identified peptides were then validated with IFN-γ ELISpot assays using PBMCs of women who have the specific HLA class I alleles. Epitope-specific CD8+ T cells were further phenotyped for memory markers with tetramer staining [45]. The iTopia Epitope Discovery System is an in vitro CD8+ T cell epitope screen system. The system consists of three assays to identify and characterize T cell epitopes: peptide binding assay, off-rate assay, and affinity assay [69,70]. Overlapping peptides spanning the Gag protein of HIV-1 subtype A and D (predominant HIV subtypes in Kenya) consensus sequences were synthesized (JPT Peptides Technologies, Inc.). The HIV Gag peptide library consists of 632 peptides (9-mer overlapping by 8 amino acids) incorporating sequence variations in the subtype A and D consensus. Of the 632 overlapping peptides screened with iTopia Epitope Discovery system, A*01:01 only binds to three Gag peptides [45]. Among them, there is only one peptide (NSSKVSQNY) with good binding to A*01:01. Further screening of the viral variants of this peptide showed that all peptides bind to A*01:01 equally well or better [45], whereas B*07:02 binds well to 30 Gag peptides spanning the entire HIV Gag [45].

The results were not exactly as expected. We did expect that there would be differences in the Gag peptide binding specificity between A*01:01 and B*07:02 alleles, but did not expect that A*01:01, the allele associated with protection and slower seroconversion, would recognize only three Gag epitopes, while B*07:02, the allele associated with rapid seroconversion could bind to 30 Gag epitopes. IFN-γ ELISpot assay with the PBMCs of sex workers confirmed the results of the iTopia Epitope Discovery system [45]. Thus, the major difference in HIV Gag epitopes between A*01:01 and B*07:02 is the spectrum of Gag epitopes recognized, not the peptide binding affinity, off-rate, the locations of the epitopes, or epitope-specific Tem/Tcm frequencies [45]. The results showed that recognizing more HIV Gag epitopes is not better for the prevention of HIV infection because B*07:02 recognizes 10-fold more HIV Gag epitopes than A*01:01, generates strong IFN-γ ELISPOT responses, and is associated with increased susceptibility to HIV acquisition [45].

Because infection of CD4+ T cells is the key difference between HIV and other infectious pathogens, and HIV propagates better in the activated CD4+ T cells, a narrower spectrum of HIV Gag epitope recognition by a protective HLA class I allele appears to make sense. Theoretically, recognizing more epitopes would activate more CD8+ T cells to destroy the viral infected cells. However, it could also activate more CD4+ T cells via cytokines. The increased CD4+ T cell activation and recruitment to mucosal sites have the potential to enhance HIV acquisition. This could explain why B*07:02, an allele that can recognize a broad spectrum of Gag epitopes, is associated with rapid seroconversion. Ideally, an effective preventative vaccine for HIV should be able to destroy the viral infected cells without causing excessive immune activation. The narrow and focused Gag epitope presentation by A*01:01 might provide such balance, enabling the destruction of initially infected cells with minimum immune activation. Furthermore, studies have shown that, in most cases, the mucosal acquisition of HIV typically results from a single or a few founder viruses [71,72]. Immune mechanisms preventing the establishment of a few founder viruses would likely be different from those dealing with a full-blown viral infection after the virus has been well established in the host.

### 3.3. From HLA Epitope Analysis to Novel HIV Vaccine Development and Testing

It is possible that a low magnitude, narrowly focused, well maintained, virus-specific CD8+ T cell response to multiple subtypes is sufficient to destroy and eliminate a few founder viruses without inducing inflammatory responses that may activate more CD4+ T cells and provide more targets for HIV. Can we achieve this with a vaccine? What should be the target? The A*01:01 HIV Gag epitope provided a clue. The only HIV Gag peptide recognized by A*01:01 with relative high affinity and a normal off-rate is a 9-mer peptide alongside the protease cleavage site at p17/p24 [45]. This region is relatively conserved among major HIV subtypes (A1, B, D, and G) [73]. Why is this region important for HIV? The protease of HIV is a small 99-amino acid aspartic enzyme that mediates the cleavage of Gag, Gag-Pol and Nef precursor polyproteins [74]. The process is highly specific, temporally regulated and essential for the production of infectious viral particles [75,76,77,78,79,80]. Because a total of twelve proteolytic reactions are required to generate a viable virion, a vaccine generating immune responses to the sequences surrounding the 12 protease cleavage sites of HIV might be able to destroy virus infected cells, drive viral mutations to generate non-infectious viral progenies and take advantage of the rapid mutations of HIV. Therefore, a vaccine targeting sequences surrounding the 12 protease cleavage sites (PCS) could be effective [81]. Furthermore, since the sequences surrounding the 12 PCSs are highly conserved among major HIV subtypes, direct immune responses against the PCSs would yield several major advantages [73]. First, the host immune response could destroy the viral infected cells before the virus can establish itself permanently in the host. Second, the vaccine could force the virus to accumulate mutations, eliminating the normal function of the HIV protease and thus eliminating infectious virions. Third, restricting the immune responses to these sites can avoid distracting immune responses that often generate unwanted inflammatory responses and excessive immune activation, leading to more targets for HIV infection, establishment and spread. A vaccine focusing on the sequences around the 12 PCS of HIV is like a surgical attack of the HIV protease function with 12 sets of ammunition, in the meantime, minimizing the level of mucosal T cell activation that has been proposed as a critical factor in developing an effective mucosal AIDS vaccine [82]. Vaccines generating immune responses against the 12 substrates of HIV protease can present a high barrier for virus to escape while avoiding the side effect of excessive T cell activation.

Sequences surrounding the HIV PCSs are immunogenic and highly conserved across global HIV subtypes when compared with Gag and Pol sequences outside these regions. Their conservation is comparable to the sequences of other identified functional motifs in Gag and the sequences targeted by reverse-transcriptase inhibitors NRTIs and NNRTIs [73]. Thus, a vaccine focusing immune responses on sequences surrounding these sites should be effective across multiple HIV subtypes. Our studies also showed that many HLA class I alleles common among distinct populations worldwide, restrict validated epitopes surrounding multiple PCSs [73,83]. Based on the validated CD8 T cell epitopes surrounding the HIV PCSs, an HIV vaccine targeting sequences surrounding the PCSs can cover 97.8% of the world’s population with average epitope hits of 13.68 [83]. The population coverage and average epitope hits are very high for North American (98.36%, 14.26), European (99.12%, 15.82), African (South (95.53%, 7.77), East (96.60%, 8.73), West (94.13%, 9.14), Central (93.70%, 7.76) and North (95.11%, 9.33)), and Asian (>90%, ~10.1) populations [83]. Therefore, a vaccine targeting sequences surrounding the 12 PCSs can potentially generate CD8 T cell responses to eight or all the PCSs in a given individual. In addition, T cell responses [73] and antibodies targeting PCSs (data not shown) have also been detected in Kenyan female sex workers including HIV-resistant sex workers. Thus, a vaccine against the 12 HIV PCSs should generate both T cell and antibody responses against multiple sites in most populations worldwide.

With funding from NIH’s “Innovation for HIV Vaccine Discovery” program and CIHR bridge fund, we tested the PCS vaccine approach in a nonhuman primate/SIV model [84]. Our preclinical studies demonstrated that an SIV vaccine targeting the sequences surrounding the 12 PCSs is protective. Among the nonhuman primates, Rhesus monkeys are the more commonly used animal model for HIV vaccine studies in North America, and Cynomolgus macaques are used more often in Europe [85]. Both rhesus and cynomolgus monkeys are old world monkeys. They are susceptible to SIV infection and have a natural course of SIV infection leading to simian AIDS that recapitulates HIV infection in humans. Thus, they are valid vaccine models for the evaluation of candidate HIV vaccines. Each of these NHP species presents advantages and disadvantages [86] for evaluating candidate vaccines, especially using female monkeys to model vaginal HIV infections. The menstrual cycles of Cynomolgus monkeys are similar to humans, while rhesus monkey’s menstrual cycles are seasonal [86]. We chose female Cynomolgus macaques for the vaccine study because their menstrual cycles are similar to humans, and chose SIVmac251 as the challenge virus because the SIVmac251 is a swamp of pathogenic SIV viruses, and probably the most stringent virus to date among those which more closely represent naturally transmitted isolates of HIV-1 in terms of resistance to neutralizing antibodies, co-receptor CCR5 usage, preferential replication in memory CD4+T cells and progression to AIDS (see review [85]).

In the study, female Mauritian Cynomolgus macaques (MCMs) were vaccinated with the PCS vaccine based on the sequences surrounding the PCS of SIVmac239. The PCS vaccine delivers 12 20-amino acid peptides using a combination of two vaccine platforms, a modified recombinant vesicular stomatitis virus (rVSVpcs) and nanoformulations (NANOpcs). After immunizations with the PCS vaccine, female MCMs were challenged repeatedly with pathogenic SIVmac251 by the intravaginal route. The data of the study showed for the first time that a vaccine targeting sequences surrounding PCS provided significant protection against repeated pathogenic SIVmac251 intravaginal challenges [84]. Notably, the PCS vaccine protected the monkeys without full-length Gag and Env immunogens.

### 3.4. The T Cell Responses Generated by Immunization with the PCS Vaccine Correlated with Vaccine Efficacy

We assessed monkeys’ vaginal mucosal inflammatory cytokines, vaginal mucosal and plasma antibodies, PBMC cytokine-producing antigen recall responses and CD4+ or CD8+ T cell subsets (memory, IL-17A+ and Treg, ex vivo or Ag recall) at the peak (week 73, one week after the last immunization) and pre-challenge (week 90) time points. All seven immune correlates of protection are T cells. These include RANTES, MIP-1α and IL-6 secreted by PBMCs after PCS peptide stimulation at the peak time point (week 73), and the frequency of the PCS peptide-specific Treg effector memory cell population at the pre-challenge time point (week 90) [84]. In addition, without antigen stimulation, the frequency of CCR5 expressing Th17 TEM cells and the frequency and intensity of CCR5 expressing CD8+IL17A+ TEM cells are also correlated with vaccine efficacy [84]. The effect of PCS peptide-specific T cell responses on vaccine efficacy was clearly shown by a multiple regression analysis of the immunological data using LASSO [84]. The LASSO regression model identified six predictors that accurately predicted vaccine efficacy. All six immune predictors are PCS peptide-specific T cells. Of them four are PCS peptide-stimulated Treg cells. The analysis showed that these antigen-specific Treg cells acted together with PCS peptide-specific IL2+CD8+T central memory cells and PCS peptide-specific CD107a+IL17+ CD8+T effector memory cells to regulate the immunological microenvironment in the vaccinated macaques and contribute to the effective protection against pathogenic SIVmac251 acquisition [84]. Thus, immunization with the PCS vaccine generated an immunological microenvironment with higher frequencies of PCS peptide-specific CD8+ central memory T cells that have better potential to kill viral infected cells and a higher frequency of PCS peptide-specific T regulatory cells (Treg) and a higher frequency of naïve Treg expressing IL-10, as well as fewer viral target cells [84].

The immune microenvironment generated by immunizations with the PCS vaccine is similar to what has been observed in the HIV-resistant female sex workers. The PCS peptide-specific CD8+ T cell responses mimic the narrow spectrum and lower magnitude CD8+ T cell responses observed in HIV-resistant sex workers [27,28]. Similar to the lower level of CD4 T cell activation and the Treg cells observed in the HIV-resistant female sex workers [42,87], the PCS peptide-specific Treg cells play an important role in the vaccine efficacy [84]. The study also showed that RANTES, MIP-1α and IL-6 secreted by the PCS peptide-stimulated CD8 T+ cells are protective, but not IFN-γ [84].

Currently, Malaika Vx, a start-up company is working to bring the PCS vaccine to clinical trials.

## 4. Role of FREM1 and Its Isoform TILRR in HIV-1 Acquisition

### 4.1. A Low-Resolution Single Nucleotide Polymorphism (SNP) Analysis Identified a FREM1 SNP rs1552896

The roles of HLA class I and class II genes in the induction and regulation of the immune responses are well recognized. Thus, it is logical to study the role of HLA class I and class II genes in the natural immunity observed in the HIV-resistant women of the Pumwani cohort. However, polymorphism of other genes not known to be involved in immune responses may also play a role in the natural immunity to HIV in these women. Genome-wide SNP (single nucleotide polymorphism) analysis allows for an unbiased identification of genetic factors that influence complex traits, and high-throughput technologies have allowed for tremendous progress in this field [88,89,90]. With the funding of the Bill and Melinda Gates foundation and CIHR, we conducted a low-resolution genome-wide SNP analysis of a subset of female sex workers [46]. The analysis identified the minor allele of an SNP rs1552896 was significantly enriched in HIV-resistant women [46]. To confirm the association we genotyped the entire sex worker cohort for this SNP by PCR and Sanger sequencing. To validate the finding we genotyped this SNP of a second Kenyan cohort. The minor allele of the SNP rs1552896 was enriched in HIV-resistant sex workers and also in uninfected women of a mother–child HIV transmission cohort [46].

The SNP rs1552896 is located in an intron of FREM1 (FRAS1-related extracellular matrix protein 1); a gene was identified in humans in 2006 as the mouse frem1 gene equivalent [91,92]. FREM1 is an extracellular matrix protein that plays a critical role in epithelial–mesenchymal interactions [92,93,94,95]. Mutations of FREM1 are associated with multiple phenotypic abnormalities [91,96]. At the time there were few studies about FREM1, let alone its role in HIV infection. Because FREM1 is an extracellular matrix protein, it may express in mucosal tissues that are relevant to HIV infection. We examined the FREM1 mRNA expression in different human tissues and found that FREM1 mRNA is highly expressed in mucosal tissues, especially the cervix, small intestine and colon [46]. Immunohistochemical staining of ectocervical tissue of a selected HIV-resistant woman from the Pumwani sex worker cohort located FREM1 proteins in both the epithelial layer and upper lamina propria [46]. Thus, FREM1 is highly expressed in tissues relevant for HIV-1 infection; further investigation is warranted.

### 4.2. The Potential Role of FREM1 in HIV Infection

Among the few publications related to FREM1 at the time, a paper by Dr. Qwarnstrom’s lab provided a clue about the possible role FREM1 may play in HIV infection. The paper showed that a heparin binding protein that had been studied for many years is actually a splice variant of FREM1 [97]. The group named the protein as TILRR (Toll-like/Interleukin 1 Receptor Regulator) because it is shown to associate with IL-1/IL-1R1 and potentiates the activation of NF-κB transcription factor and inflammatory responses [97]. Several alternative splice variants of FREM1 have been identified; however, TILRR was the only one studied [98]. To study the role of FREM1 in HIV infection, we conducted RNA-seq analysis of the PBMCs of Kenyan female sex workers. TILRR mRNA was detected in all women who are homozygous for the major allele of rs1552896. However, the TILRR mRNA is either absent or expressed at a very low level in women with the minor allele of rs1552896 [98]. Because TILRR is shown to potentiate MyD88 recruitment to control Ras-dependent amplification of NF-κB, we hypothesized that the association of the rs1552896 minor allele with resistance to HIV infection may be partly due to an impaired IL-1R1/TLR signal transduction resulting from the abolition or reduced TILRR expression and reduced inflammatory responses.

Our subsequent studies showed that TILRR is an important modulator of many genes involved in the inflammatory responses and promotes inflammatory cytokine secretion by epithelial cells [99]. TILRR promotes immune cell migration through the induction of soluble inflammatory mediators [100]. More recently we discovered that TILRR protein is not only expressed in PBMCs and tissues, but also circulates in blood [101]. The levels of plasma TILRR protein among individuals vary greatly, ranging from less than 2.38 ng/mL to >5 µg/mL [101], and the levels of plasma TILRR protein were positively correlated with several proinflammatory cytokines in the blood plasma [102]. These observations are consistent with our in vitro studies showing that TILRR promoted the secretion of several proinflammatory cytokines [99,100]. The presence of TILRR protein, an inflammatory response modulator, in blood plasma, and its correlation with levels of several proinflammatory cytokines, suggest that TILRR not only modulates inflammatory responses in cells and tissues but may also modulate systemic inflammation. The great variations in the levels of plasma TILRR protein among individuals and its positive correlations with several proinflammatory cytokines suggest that individuals with a high level of plasma TILRR protein could be at a higher risk of HIV infection. Our studies showed that this is indeed the case [102]. We analyzed the plasma TILRR protein of 941 archived HIV negative plasma samples of 390 women who were HIV negative when they were enrolled in the Pumwani Sex Worker cohort. The influence of the levels of plasma TILRR protein on HIV seroconversion was analyzed using Kaplan–Meier survival analysis. We found that the high levels of plasma TILRR protein are associated with faster HIV seroconversion (*p* < 0.00001) [102]. Thus, the high level of plasma TILRR protein is a high risk factor for HIV acquisition, even though vaginal HIV infection is the route of HIV-1 transmission in the Pumwani sex worker cohort.

It has been more than a decade since the identification of an SNP rs1552896 in the FREM1 gene by a low-resolution, nonbiased genome-wide SNP analysis of the HIV-resistant female sex workers. Since then we have developed monoclonal antibodies to study their potential functional relevance to HIV infection [103], and we have developed a hypothesis and studied the variant protein TILRR’s role in modulating inflammatory-related genes and proinflammatory cytokines [99], and in promoting the migration of immune cells [100]. We discovered that TILRR protein is circulating in blood [101], and showed that a high level of plasma TILRR protein is a risk factor to HIV acquisition [102]. The identification of high blood TILRR protein as a risk factor for systemic inflammation and HIV acquisition further demonstrated that systemic inflammation is a risk factor for HIV acquisition, and plasma TILRR protein could be a target for reducing systemic inflammation and for HIV prevention.

## Figures and Tables

**Figure 1 viruses-14-01243-f001:**
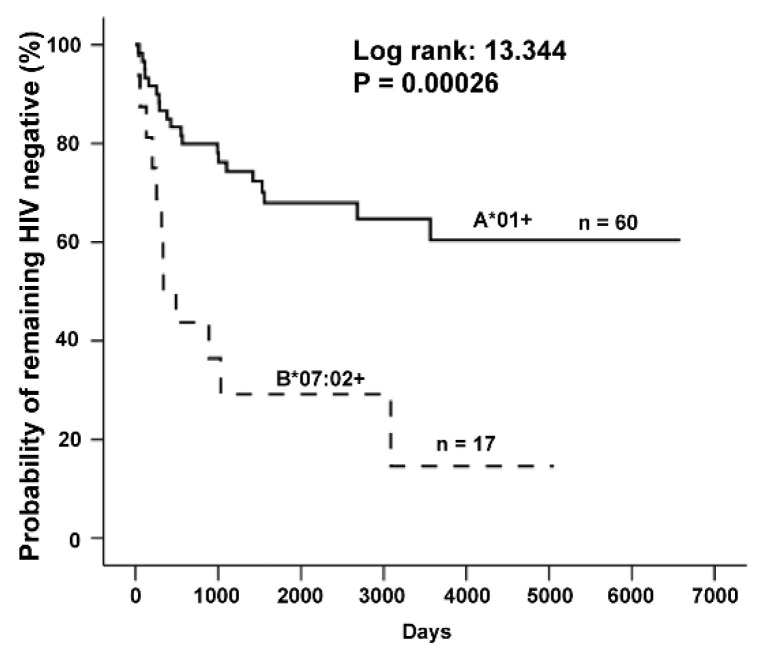
Kaplan-Meier plot of time to HIV-1 seroconversion. Comparison of A*01+ and B*07:02+ female sex workers.

## Data Availability

The data of all paper cited in this review can be found in the cited published paper.

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
