# Peer review of "Natural Immunity against HIV-1: Progression of Understanding after Association Studies"

_viruses, 2022, doi:10.3390/v14061243_

Round 1

Reviewer 1 Report

This is a very eloquent and thoughtful review of innovative reports and providing some of the first encouraging data moving towards an HIV vaccine. The findings are also consistent with some of this reviewer's unpublished observations that individuals with low levels of CD4 activation and correlating with HLA types of low responses also had low viral loads. The lack of correlation between vaccine protection and IFNg is also of interest, as is the section on TILRR.

Author Response

A point-by-point response to the reviewer’s comments is as follows:

Reviewer 1:  

This is a very eloquent and thoughtful review of innovative reports and providing some of the first encouraging data moving towards an HIV vaccine. The findings are also consistent with some of this reviewer's unpublished observations that individuals with low levels of CD4 activation and correlating with HLA types of low responses also had low viral loads. The lack of correlation between vaccine protection and IFNg is also of interest, as is the section on TILRR.”

My Response: I thank the reviewer for the comments. In addition, the revision has also been checked for style and language by an expert with excellent English language skills as suggested by the reviewer.

Reviewer 2 Report

The manuscript by Dr. Luo reviews the natural immunity to HIV-1 using association studies to guide in the development of potential vaccine against this virus.  The review discusses the HLA class I genes of sex workers who become infected with HIV-1 (e.g., B*07:02) versus those who are more resistant (A*01:01) against infection. Using this approach, they  identify peptides overlapping protease cleavage sites as a potential vaccine strategy.  Overall, the manuscript is well written will provide a valuable information on the use of association studies to identify potential vaccine candidates.

Major comments:

1. The author discusses the use of a vaccine with a series of peptides corresponding to 12 cleavage sites within the SIVmac239 genome. Following repeated vaccinations, the authors challenged these Mauritian cynomolgus macaques with SIVmac251 using low dose challenges and concluded that targeting protease cleavage sites is a novel vaccine approach. However, cynomolgus macaques are not the most stringent for SIV vaccine studies.  The author should discuss rationale for the use cynomolgus macaques instead of rhesus macaques and SIVmac251 instead of SIVmc239 in their studies.

Minor comments:

1. Line 11. Delete the duplicated “the.”

2. Line 21. In the sentence “worked well for other infectious pathogens failed for HIV-1?” should be changed to “worked well for other infectious pathogens but failed for HIV-1?”

3. Line 140. In the sentence “dominant HIV subtypes in Kenya) consensus were synthesized” should be changed to “dominant HIV subtypes in Kenya) consensus sequences were synthesized.”

4. Lines 107 to 122. The x-axis on the graph has no label (at least not on my copy).

Author Response

Reviewer 2:

The manuscript by Dr. Luo reviews the natural immunity to HIV-1 using association studies to guide in the development of potential vaccine against this virus.  The review discusses the HLA class I genes of sex workers who become infected with HIV-1 (e.g., B*07:02) versus those who are more resistant (A*01:01) against infection. Using this approach, they  identify peptides overlapping protease cleavage sites as a potential vaccine strategy.  Overall, the manuscript is well written will provide a valuable information on the use of association studies to identify potential vaccine candidates.”

My response: I thank the reviewer for the comments.

Major comments:

The author discusses the use of a vaccine with a series of peptides corresponding to 12 cleavage sites within the SIVmac239 genome. Following repeated vaccinations, the authors challenged these Mauritian cynomolgus macaques with SIVmac251 using low dose challenges and concluded that targeting protease cleavage sites is a novel vaccine approach. However, cynomolgus macaques are not the most stringent for SIV vaccine studies.  The author should discuss rationale for the use cynomolgus macaques instead of rhesus macaques and SIVmac251 instead of SIVmc239 in their studies.”

My response: In response to the reviewer’s comments I have added a section to the review article as follows:

“Among the nonhuman primates, Rhesus monkeys are the more commonly used animal model for HIV vaccine studies in North America, and Cynomolgus macaques are used more often in Europe[85]. Both rhesus and cynomolgus monkeys are old world monkeys. They are susceptible to SIV infection and have a natural course of SIV infection leading to simian AIDS that recapitulates HIV infection in humans. Thus, they are valid vaccine models for evaluation of candidate HIV vaccines. Each of these NHP species presents advantages and disadvantages[86] for evaluating candidate vaccines, especially using female monkeys to model vaginal HIV infections.  The menstrual cycles of Cynomolgus monkeys are similar to humans, while rhesus monkey’s menstrual cycles are seasonal[86].  We chose female Cynomolgus macaques for the vaccine study because their menstrual cycles are similar to humans, and chose SIVmac251 as the challenge virus because the SIVmac251 is a swamp of pathogenic SIV viruses, and probably the most stringent virus to date among those which more closely represent naturally transmitted isolates of HIV-1 in terms of resistance to neutralizing antibodies, co-receptor CCR5 usage, preferential replication in memory CD4+T cells, and progression to AIDS (see review[85]).

Minor comments:

  1. Line 11. Delete the duplicated “the.”

My response: Done.

  1. Line 21. In the sentence “worked well for other infectious pathogens failed for HIV-1?” should be changed to “worked well for other infectious pathogens butfailed for HIV-1?”

My response: This sentence has been revised according to the reviewer’s suggestion.

  1. Line 140. In the sentence “dominant HIV subtypes in Kenya) consensus were synthesized” should be changed to “dominant HIV subtypes in Kenya) consensus sequenceswere synthesized.”

My response: Done as suggested.

  1. Lines 107 to 122. The x-axis on the graph has no label (at least not on my copy).

My response: The X-axis label was covered by the box during file transformation into Viruses format. Adjustment has been made in the revision.
